# Routine Screening for Central and Primary Adrenal Insufficiency during Immune-Checkpoint Inhibitor Therapy: An Endocrinology Perspective for Oncologists

Irena Druce [1,*], Karine Tawagi [2], Julie L. V. Shaw [3], Andrea Ibrahim [2], Heather Lochnan [1] and Michael Ong [2]

[1] Division of Endocrinology and Metabolism, Department of Medicine, The Ottawa Hospital Research Institute, University of Ottawa, Ottawa, ON KIH8L6, Canada; hlochnan@toh.ca

[2] Division of Medical Oncology, Department of Medicine, The Ottawa Hospital Research Institute, University of Ottawa, Ottawa, ON KIH8L6, Canada; karine.tawagi@gmail.com (K.T.); ibrahim.andrea@gmail.com (A.I.); mong@toh.ca (M.O.)

[3] Department of Pathology and Laboratory Medicine, Eastern Ontario Regional Laboratories Association, The Ottawa Hospital Research Institute, The Ottawa Hospital, The University of Ottawa, Ottawa, ON KIH8L6, Canada; julshaw@eorla.ca

\* Correspondence: idruce@toh.ca

**Abstract:** Background: Immune checkpoint inhibitor (ICI)-associated hypothalamic–pituitary–adrenal axis disruption can lead to hypocortisolism. This is a life-threatening but difficult to diagnose condition, due to its non-specific symptoms that overlap with symptoms of malignancy. Currently, there is no consensus on how to best screen asymptomatic patients on ICI therapy for hypophysitis with serum cortisol. Methods: A retrospective chart review of patients treated with ICI in a tertiary care centre was conducted to assess the rate of screening with cortisol and whether this had an impact on diagnosis of ICI-hypophysitis in the preclinical stage. Patients were identified as having hypophysitis with an adrenocorticotropin hormone (ACTH) deficiency based on chart review of patients with cortisol values ≤ 140 nmol/L (≤5 mcg/dL). We also assessed what proportion of cortisol values were drawn at the correct time for interpretation (between 6 AM and 10 AM). Results: Two hundred and sixty-five patients had 1301 cortisol levels drawn, only 40% of which were drawn correctly (between 6 and 10 AM). Twenty-two cases of hypophysitis manifesting with ACTH deficiency were identified. Eight of these patients were being screened with cortisol following treatment and were detected in the outpatient setting. The remaining 14 patients were not screened and were diagnosed when symptomatic, after an emergency room visit or hospital admission. Sixty percent of the cortisol tests were uninterpretable as they were not drawn within the appropriate time window. Conclusion: Measuring morning serum cortisol in asymptomatic patients on ICI therapy is a fast and inexpensive way to screen for hypophysitis and should become the standard of care. Random serum cortisol measurement has no clinical value. Education needs to be provided on when to correctly perform the test and how to interpret it and we provide an algorithm for this purpose. The adoption and validation of such an algorithm as part of routine practice could significantly reduce morbidity and mortality in patients, especially as ICI therapy is becoming increasingly commonplace.

**Keywords:** immune checkpoint inhibitor; hypophysitis; screening

## 1. Introduction

Immune-checkpoint inhibitors (ICIs) targeting cytotoxic T-lymphocyte-associated protein-4 (CTLA-4; ipilimumab and tremelimumab), programmed-death receptor-1 (PD-1; nivolumab, pembrolizumab and cemiplimab) or its ligand, PD-L1 (atezolizumab, durvalumab and avelumab), have been approved for the treatment of a variety of malignancies [1–10]. There are over a thousand clinical trials which involve ICIs, alone or in combination with other therapies, and the number of patients exposed to these agents is

rising rapidly [11]. It is well-established that ICIs can result in immune-related adverse events (irAEs). A systematic review found that the rates of any irAE with anti-PD-1/PD-L1 inhibitors were 74% (14% grade ≥ 3), with anti-CTLA-4 were 89% (34% grade ≥ 3), and with anti-PD-1 + anti-CTLA-4 were 90% (55% grade ≥ 3) [12]. These irAEs can range from very mild to life-threatening and have been shown to manifest in virtually every organ system of the body.

ICI-related irAEs of the endocrine system are common, although the clinical manifestations and implications of thyroid, pituitary, pancreatic or adrenal toxicity are quite different. For example, ICI-associated thyroid dysfunction manifests most often as subclinical thyroiditis. After a period of hyperthyroidism, the thyroiditis will often self-resolve without treatment, or can progress to permanent hypothyroidism, which is easily diagnosed biochemically and generally easily treated with thyroid hormone replacement. In contrast, ICI-associated hypophysitis and ICI-associated primary adrenal insufficiency lead to a permanent loss of adrenal function and a potential clinical presentation with vasodilatory collapse and adrenal crisis, which can be grossly underrecognized [7]. The difficulty in recognition of ICI-associated hypothalamic–pituitary–adrenal (HPA) axis disruption (ICI-HPA) is that it presents with non-specific symptoms, which can easily be mistaken for a myriad of other conditions. Manifestations of fatigue, weakness, poor oral intake, low blood pressure, nausea and vomiting [13] are all symptoms which are commonly encountered in patients with advanced malignancy receiving systemic therapy.

Screening and monitoring for the development of endocrine irAEs are therefore of high importance. In particular, the HPA axis alterations have the potential to produce severe and rapid-onset manifestations, because the half-life of adrenocorticotropic hormone (ACTH) and cortisol is in the order of minutes, and the downstream effects of this axis are dynamic and acutely felt. In a time of physical stress that is common with cancer patients, a deficiency of adrenal hormone can manifest as a life-threatening adrenal crisis [14]. This can be contrasted with measurement and monitoring of thyroid hormones, such as thyroid-stimulating hormone (TSH), which has a relatively long half-life, measured in days, and regulates more indolent processes in the body, such as basal metabolic rate and energy expenditure.

However, screening for ICI-HPA axis disruption presents several important challenges; the incidence is low, presentation is insidious, biochemical screening is time-sensitive and confounded by the use of exogenous steroids, and consensus recommendations are lacking.

This paper explores the ICI-HPA screening practices at a tertiary-care institution and whether these practices had a clinical impact. All laboratory testing of cortisol was extracted and characterized for time of blood draw, concurrent systemic steroid use, presence or absence of clinical symptoms and/or other endocrinologic syndrome. Based on these findings, a testing algorithm for the proper procedure and test interpretation of serum cortisol is proposed.

## 2. Methods

A retrospective chart review was conducted of all patients who received a dose of ICI at The Ottawa Hospital (TOH)—a multi-campus academic teaching hospital located in Ottawa, Ontario that serves a catchment area of 1.3 million people. Patients who had received at least one dose of an ICI agent, both in a trial and non-trial setting, between 1 January 2014 to 31 December 2018, were included in the cohort. This study was conducted in accordance with The Ottawa Hospital Research Ethics Board's protocol for chart reviews (Protocol ID 5682, approved 27 October 2016).

### 2.1. Cortisol Measurement

Serum cortisol levels drawn in the patient cohort were extracted from the laboratory information system at TOH. Serum cortisol was measured by immunoassay on the Beckman Unicell DXI 800 instrument (Beckman Coulter, Brea, CA, USA). Patients within the ICI cohort who had a cortisol of ≤140 nmol/L (≤5 mcg/dL), based on criteria used to define

adrenal insufficiency [15], were identified. A detailed chart review was used to determine the etiology for the low cortisol value, and the cortisol result was categorized as due to hypophysitis, concurrent systemic glucocorticoid use, pre-existing HPA disease or insufficient data. For cortisol levels which were categorized as normal, we still assessed which proportion of them were drawn within the appropriate window (between 6 and 10 AM) to determine which were being drawn correctly for the purpose of identifying HPA axis dysfunction. Patient demographic information was collected including the agent used, the start and stop of ICI therapy, sex, age, malignancy site, whether the ICI was used with adjuvant or palliative intent and whether it was part of a clinical trial.

### 2.2. Immune-Checkpoint-Inhibitor-Associated HPA-Axis Dysfunction

Patients in the cohort with a clinically significant ICI-HPA were identified based by reviewing patients in the low-cortisol category and determining whether they had been referred to Endocrinology. Patients with referrals for other presenting complaints were excluded, for example if the patient had new-onset diabetes. Demographic data as well as the location of initial presentation, the timing of onset and which ICI class the patient had received was collected. Not all patients had ACTH measured, which would allow for definitive discrimination between primary and central adrenal insufficiency. For those patients without ACTH data, the classification between primary and central was made based on whether the patients were prescribed fludrocortisone.

### 3. Results

#### 3.1. Patient Cohort

Seven hundred and eighty-six patients received a dose of ICI in the specified time-period. Of those patients, 265 patients had a cortisol level drawn at TOH following receipt of at least once cycle of ICI (Table 1). The median age of patients was 66 years. The underlying malignancy was melanoma ($n$ = 124, 46.8%), lung cancer ($n$ = 66, 24.9%), and genitourinary cancer ($n$ = 41, 15.5%). In terms of treatment type, most patients were on PD1/PDL1 monotherapy ($n$ = 168, 63.4%), with the remainder on CTLA4 monotherapy ($n$ = 49, 18.5%) or CTLA4 in combination with PD1/L1 therapy ($n$ = 48, 18.1%). There were 83 patients (31.4%) who received an ICI as part of a clinical trial.

**Table 1.** Demographics of patients receiving ICI who had cortisol levels drawn.

|  |  | **Number** | **Percent** |
|---|---|---|---|
| Gender (n = 265) |  |  |  |
|  | Male | 178 | 67.42 |
|  | Female | 86 | 32.58 |
| Age (years, median) |  | 65.5 |  |
| Disease site ($n$ = 265) |  |  |  |
|  | Melanoma | 124 | 46.8 |
|  | Thoracic | 66 | 24.9 |
|  | Genitourinary | 41 | 15.5 |
|  | Gastrointestinal | 11 | 4.2 |
|  | Head & Neck | 9 | 3.4 |
|  | Other | 14 | 5.3 |
| Treatment type ($n$ = 265) |  |  |  |
|  | PD-1/PDL-1 monotherapy | 168 | 63.4 |
|  | CTLA-4 monotherapy | 49 | 18.5 |
|  | CTLA-4 combination with PD-1/PDL-1 | 48 | 18.1 |
| Intent of Treatment ($n$ = 265) |  |  |  |
|  | Palliative | 256 | 96.59 |
|  | Adjuvant | 9 | 3.41 |
| Clinical Trial ($n$ = 265) |  |  |  |
|  | Yes | 83 | 31.42 |
|  | No | 182 | 68.67 |

*3.2. Cortisol Testing*

1301 serum cortisol levels were drawn in the study population, corresponding to 265 individual patients. It is likely that the figure of only 265/786 patients having a cortisol measured is an under-representation as only cortisol measurements performed at our institution were captured. Many patients chose to get their routine blood work done at community laboratories, and these data were not collected. The decision to measure cortisol routinely was at the treating physician's discretion and may be another reason for the low number of tests.

The median number of serum cortisol levels drawn per patient within the four-year study period was two, with a range of 1–56. Using a cut-off of $\leq 140$ nmol/L ($\leq 5$ mcg/dL), 226 (17%) values fell below the detection limit, corresponding 76 (29%) patients. The etiology for the low cortisol is summarized in Table 2.

**Table 2.** Etiology of low serum cortisol values.

| Etiology | Cortisol $\leq$ 140 nmol/L ($\leq$5 mcg/dL) *n* (%) | Method of Diagnosis |
|---|---|---|
| Concurrent systemic glucocorticoid use | 36 (47.4) | Medical record included active prescription for systemic glucocorticoid at time of blood draw |
| Hypophysitis | 22 (28.9) | Patient went on to require glucocorticoid replacement and Endocrinology referral |
| Pre-existing HPA dysfunction | 2 (2.6) | Past medical history included pre-existing etiology of HPA dysfunction and patient was on glucocorticoid therapy prior to ICI administration |
| Data uninterpretable | 16 (21.1) | No record of concurrent glucocorticoid use; patient did not go on to require glucocorticoid replacement |

Twenty-two patients with HPA-axis pathology were identified. Two patients had pre-existing dysfunction of their HPA axis (one case of Addison's disease and one case of pituitary apoplexy) to explain their low serum cortisol. The majority (47.4%) of patients had a low cortisol value due to concurrent systemic glucocorticoid use. The remaining 16 (21.1%) patients did not have an obvious etiology for their low serum cortisol identified based on chart review; it is possible that they may have had concurrent glucocorticoid use which was not properly documented or were using topical or inhaled glucocorticoids. None of these patients had HPA axis pathology.

The serum cortisol measurements were quantified based on the time the test was drawn and the data are summarized in Figure 1. Highlighted in the figure is the cortisol levels which were drawn at the correct time (6:00 AM to 9:59 AM in this analysis) and comprises 519 (40%) of the tests. The remainder of the tests, whether the result is normal or low, are not interpretable in the ambulatory care setting.

*3.3. Immune-Checkpoint-Inhibitor-Associated HPA-Axis Pathology*

Amongst the 22 patients with HPA-axis pathology identified (Table 3), all the patients had central adrenal insufficiency and/or ACTH-deficiency. No cases of primary adrenal insufficiency were identified in our cohort. Of the 22 with ICI-HPA, eight patients had routine cortisol measurement as part of their care prior to developing ICI-HPA and were detected to have HPA-axis pathology in the outpatient setting. All the patients having regular screening were started on glucocorticoid replacement therapy within one week of the first biochemical evidence of ICI-HPA. Six of the patients were started on treatment the very same day as their blood test, while the other two patients were started at one and six days after screening.

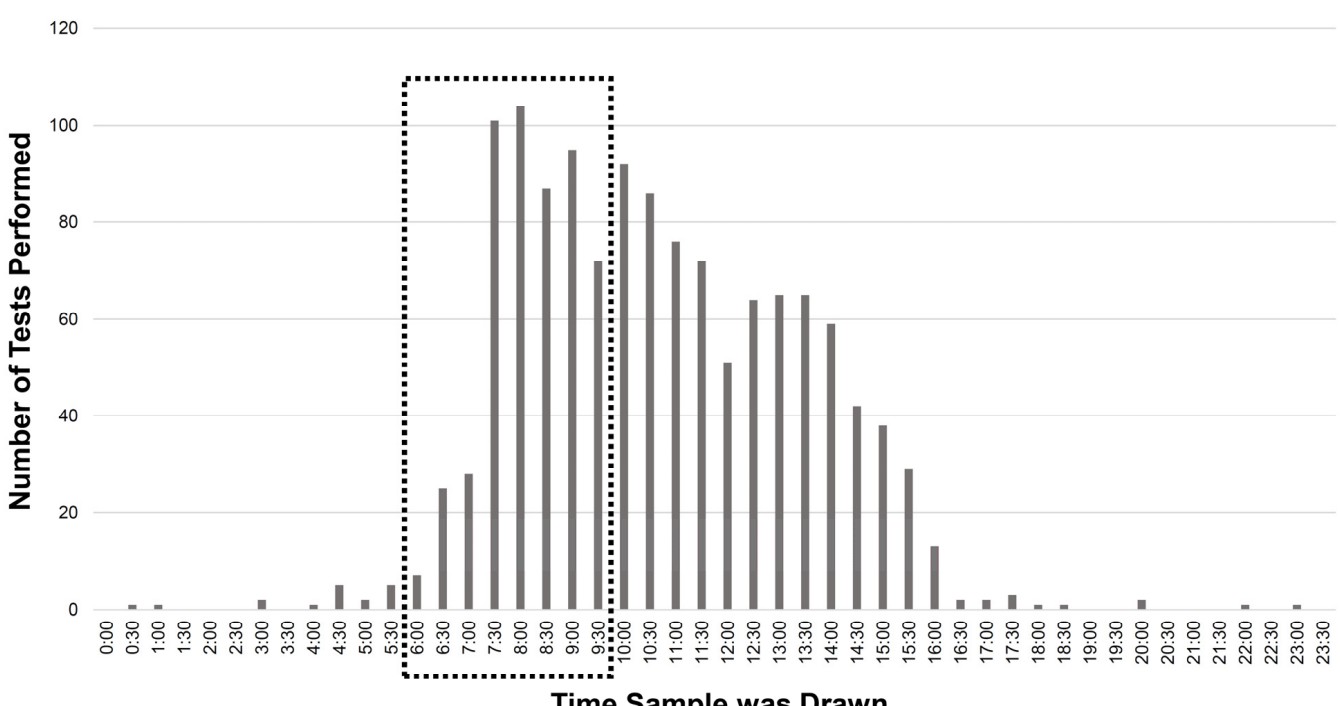

**Figure 1.** Number of serum cortisol tests performed by time in the sample population. The black rectangle highlights the tests which were drawn at the correct time.

Fourteen patients were not being screened. Eight of those patients were diagnosed in the outpatient setting based on new symptoms consistent with hypophysitis followed by confirmatory testing, while six cases of HPA-axis pathology were only diagnosed after an emergency visit or hospital admission. In all six of these cases, the emergency visit or admission was due to symptoms of adrenal insufficiency and not for another etiology where ICI-HPA was a secondary diagnosis. All these patients had their first cortisol measurement after suspicious symptoms onset only.

Diagnosis of ICI-HPA was made a median of 108 days after treatment initiation (range 34–412). 16/22 (72.7%) patients were treated with a regimen which contained anti-CTLA4 either concurrently with anti-PD1 (3/22), followed by anti-PD1 (Sequential; 6/22) or exclusively with anti-CTLA4 (7/22). Six of the patients were treated exclusively with anti-PD1. In patients whose regimen contained anti-CTLA4, diagnosis was made a median of 101.5 days after start of treatment (range 34–371), versus a median of 245 days for patients who only received a PD1 (range 93–412). None of the patients in this cohort received PDL1 therapy. Ten of the patients (45%) were diagnosed after their treatment had been discontinued.

Based on how patients with ICI-HPA were identified, all of them had secondary adrenal deficiency (ACTH deficiency), five of them also had thyroid stimulating hormone (TSH) deficiency, and six also had central hypogonadism (LH or luteinizing hormone and FSH or follicle-stimulating hormone deficiency). Six of the patients also had prolactin deficiency (Prl), although this could be an under-representation. Post CTLA4 therapy, deficiency in all pituitary hormones has been documented. Patients who develop ICI-HPA after exclusive therapy with PD1 tend to primarily have ACTH deficiency, although other axes can be affected, such as in patient 18 [16].

**Table 3.** Details of patients in cohort who had confirmed ICI-HPA.

| | Cortisol Screening (Yes/No) | Treatment Regimen | Primary Malignancy Site | Time of Diagnosis (Days) | Diagnosis after Treatment DC (Yes/No) | Diagnosed in Outpatient Setting (Yes/No) | Hormones Affected |
|---|---|---|---|---|---|---|---|
| 1 | Yes | CTLA4 + PD1 Concurrent | Melanoma | 110 | Yes | Yes | ACTH |
| 2 | Yes | CTLA4 + PD1 Concurrent | Melanoma | 104 | Yes | Yes | ACTH, TSH, LH, FSH |
| 3 | Yes | CTLA4 + PD1 Concurrent | Melanoma | 88 | Yes | Yes | ACTH, TSH |
| 4 | Yes | CTLA4, PD1 Sequential | Melanoma | 41 | No | Yes | ACTH, LH, FSH, Prl |
| 5 | Yes | CTLA4, PD1 Sequential | Melanoma | 371 | Yes | Yes | ACTH, TSH |
| 6 | Yes | CTLA4 | Melanoma | 106 | Yes | Yes | ACTH, TSH, LH, FSH, Prl |
| 7 | Yes | CTLA4 | GU | 161 | Yes | Yes | ACTH, TSH |
| 8 | Yes | PD1 | Melanoma | 207 | Yes | Yes | ACTH, Prl |
| 9 | No | CTLA4, PD1 Sequential | Melanoma | 74 | No | Yes | ACTH |
| 10 | No | CTLA4, PD1 Sequential | Melanoma | 70 | No | Yes | ACTH |
| 11 | No | CTLA4, PD1 Sequential | Melanoma | 205 | No | No | ACTH |
| 12 | No | CTLA4, PD1 Sequential | Melanoma | 192 | No | No | ACTH |
| 13 | No | CTLA4 | Melanoma | 171 | Yes | No | ACTH, Prl |
| 14 | No | CTLA4 | Melanoma | 56 | No | Yes | ACTH, LH, FSH, Prl |
| 15 | No | CTLA4 | Melanoma | 45 | No | Yes | ACTH |
| 16 | No | CTLA4 | Melanoma | 34 | No | No | ACTH, LH, FSH |
| 17 | No | CTLA4 | GU | 99 | Yes | No | ACTH, TSH |
| 18 | No | PD1 | GU | 412 | No | No | ACTH, TSH, LH, FSH, Prl |
| 19 | No | PD1 | GU | 287 | No | Yes | ACTH |
| 20 | No | PD1 | H&N | 154 | No | Yes | ACTH |
| 21 | No | PD1 | GI | 93 | Yes | Yes | ACTH |
| 22 | No | PD1 | Melanoma | 283 | No | Yes | ACTH, Prl |

DC = treatment discontinuation, GU = genitourinary, H&N = head and neck, GI = gastrointestinal, ACTH = adrenocorticotropic hormone, TSH = thyroid-stimulating hormone, LH = luteinizing hormone, FSH = follicle-stimulating hormone, Prl = prolactin.

## 4. Discussion

Immune-checkpoint-associated adrenal insufficiency is a rapid-onset and potentially life-threatening condition. Routine screening for ICI-hypophysitis with morning cortisol in asymptomatic patients should be the standard of care as it has the potential to prevent significant morbidity and mortality. Thus far, this has not been the recommendation or practice as outlined in the individual product monographs, [17–23], nor the majority of clinical trials [1–3,6,7,9].

In this single-institution review, we present 'real-world' data for cortisol screening in patients treated with immune-checkpoint inhibitors. We identified 22 patients diagnosed with ICI-HPA dysfunction. Of the eight of these patients who were having regular cortisol screening, all were identified on an outpatient basis in the pre-clinical stage. Amongst the remaining 14 patients, 6 presented to the emergency department and/or were hospitalized before diagnosis. In all these cases, adrenal insufficiency was the primary diagnosis, it was not an incidental finding during presentation for another complaint.

Measuring ACTH levels is not useful as a screening test as the result can only be interpreted together with a cortisol value and the assay turn-around times are in the order of days to weeks. The immunoassay for total serum cortisol is inexpensive, rapid and available readily at most institutions and commercial diagnostic labs. While it cannot distinguish primary from central hypocortisolism, it can detect ICI-HPA in the pre-clinical stage where further testing and treatment can be initiated. Serum cortisol measurement satisfies all the criteria for a good screening test [24]. There are, however, several challenges

with the measurement and interpretation of serum cortisol, which may explain its limited use as a screening tool in ICI-treated patients so far.

Unlike screening for thyroid dysfunction for example, cortisol measurement is time-sensitive, and the reference intervals provided by the laboratory do not necessarily correlate with the accepted cut-offs for diagnosis of deficiency. The reference intervals reported are often those recommended by the assay manufacturer, taken from the literature or calculated from in-house reference interval establishment studies. Reference intervals are calculated from measurements in normal, healthy individuals, which would not include patients with cortisol deficiency. To add to the complexity, there is even debate amongst the Endocrinology community as to what those accepted cut-offs are, and universal cut-offs are challenging given the differences in the immunoassays used to measure cortisol by different laboratories.

Endogenous cortisol secretion is pulsatile and highly variable, but it follows a diurnal pattern with peak secretion generally accepted to be between 6 and 9 AM [25] to 10 AM [15]. To document a deficiency in cortisol, plasma measurement must be carried out in that early hour window. There is consensus that random plasma cortisol has no utility in the diagnosis of HPA-axis dysfunction [15].

Morning serum cortisol values below 80–140 nmol/L (3–5 mcg/dL) indicate insufficiency and do not require additional testing [14], while it is generally accepted that levels of greater than 275 nmol/L (10 mcg/dL) rule insufficiency out reliably [15], although some sources are more stringent and would suggest that only morning cortisol values above 525 nmol/L (19 mcg/dL) [26] predict an adequate response to stress. With the goal of providing clear and practical guidance, a screening algorithm with defined and consistent cut-offs is recommended and it is in these authors' recommendation that it be in line with the Endocrine Society's published guidelines, which state that morning cortisol of ≤140 nmol/L (≤5 mcg/dL) is suggestive of cortisol deficiency and values ≥ 275 nmol/L (≥10 mcg/dL) rule it out [15]. Between 140 and 274 (2.9–9.9 mcg/dL), additional testing is required [15,27].

Within this study population, 60% of cortisol tests were performed outside of the accepted time frame. This was possibly due to lack of guidance or provider ignorance of the time-sensitivity of this test, but most likely was the result of patient factors. Prior studies have noted that patient adherence to routine laboratory testing decreases when there are additional instructions such as specific timing or fasting [28]. Requiring patients to have all their blood work done in the morning is bound to increase inconvenience but is necessary and will require provider and patient education.

The second important criterion for routine serum cortisol monitoring is the frequency and whether it should be treatment-specific. The incidence of ICI-associated hypophysitis varies depending on the agents used—incidences of up to 13% have been reported with CTLA-4 therapy but only up to 3% with PD-1/PD-L1 therapy [29]. Primary adrenal insufficiency has also been noted with ICI therapy, but its incidence is low, quoted in general as around 1%, regardless of treatment modality [29–31]. In our cohort, patients treated with anti-CTLA4 comprised the majority (16/97 or 16.4%) of those who developed hypophysitis, whether they are treated concurrently or sequentially with anti-PD1, or if they received anti-CTLA4 monotherapy, compared with 6/168 (3.6%) of patients receiving anti-PD1/PDL1 therapy exclusively.

In this population, diagnosis of ICI-HPA was made a median of 101.5 (range 34–371) days after start of treatment with a regimen containing CTLA4 (either alone, or in combination of PD1), versus a median of 245 (range 93–412) days for patients who only received a PD1. Ten of the patients (45%) developed ICI-HPA after treatment discontinuation. These findings are in line with other studies [31–37]. The observed difference between the rates and timing of onset relate to the underlying mechanism of ICI-associated hypophysitis, which is not yet well-understood but possibly involves the expression of CTLA4 and PD1 in the pituitary and variable rates of autoantibody production [32,38].

Based on these findings, it would seem to make sense to develop monitoring regimens which are treatment-specific with regard to the onset and frequency of testing, for example with more frequent testing early after treatment start with agents containing CTLA4. This would, however, add another level of complexity and may decrease utility. The patient population subject to the proposed intervention is already having routine blood work with every cycle and the cost of one additional test, in comparison to the cost of the treatment itself, is negligible. Therefore, we propose a single algorithm for the use of serum cortisol as routine screening for ICI-HPA for all patients receiving any form of ICI (Figure 2). This algorithm focuses on screening for ICI-HPA only but could be incorporated into other algorithms that screen for ICI-associated endocrinopathies. The inclusion of serum electrolytes (and renal function on which the interpretation of electrolyte levels is dependent) allows the user to quickly judge whether mineralocorticoid deficiency may be present, in the case of primary adrenal insufficiency.

The first portion of the figure relates to the frequency of testing. It is advised that all patients treated with ICI should have an early morning serum cortisol, electrolytes and renal function measurement before the start of treatment to rule out any pre-existing deficiency. During treatment, patients should have these parameters checked prior to every cycle of treatment for the first six months of therapy. If treatment is discontinued prior to the six-month mark, screening should still occur monthly up until six months after treatment initiation to account for a high-risk window for ICI-HPA incidence. The recommendation is to continue routine monthly assessments up until 12 months, regardless of whether treatment is ongoing due to high rates of diagnosis after treatment discontinuation. Beyond a year out from therapy, it is recommended that screening be undertaken based on clinical suspicion, as the likelihood of new onset of ICI-HPA drops off. The possibility does exist, however, and there needs to be ongoing clinical vigilance for patients who have ever received a dose of ICI.

The second portion of the algorithm relates to interpretation of the resulting cortisol and is based on accepted guidelines for HPA-axis deficiency outside of ICI [14,15,25]. The algorithm clearly outlines the actions based on the result of a 6–10 AM serum cortisol. The broader range of up to 10 AM was chosen according to the Endocrine Society consensus [15], to hopefully promote patient compliance by broadening the time window.

The cut-off values may need to be adjusted based on individual institutional practices and standards, but we believe that it is paramount to include them to provide clear guidance for ICI prescribers due to the complexity of cortisol interpretation as previously discussed. The algorithm indicates that for patient with cortisol $\leq$ 140 nmol/L ($\leq$5 mcg/dL), HPA-axis deficiency should be suspected. If it is safe to do so, ACTH should be measured for diagnostic clarity but if not possible, this should not delay treatment. The presence of hyponatremia $\pm$ hyperkalemia is also suggestive of primary adrenal insufficiency and mineralocorticoid deficiency. Treatment of possible mineralocorticoid deficiency is important acutely as failure to do so can lead to life-threatening electrolyte abnormalities, hypotension and hypovolemia [39].

The third portion of the proposed algorithm outlines the steps to interpret serum cortisol values which were drawn outside of the proscribed time window, as this invariably will happen. This portion of the algorithm stresses that if the patient is at all symptomatic or there is any clinical doubt, it is safest to treat empirically, but if the patient is well, it recommends the urgency at which the screening blood tests should be repeated at the appropriate time.

It is imperative to highlight that where HPA-axis pathology is concerned, if there is any diagnostic uncertainty, it is always safest to treat immediately and confirm diagnosis only secondarily [15]. This algorithm is intended to be used in the outpatient setting and assumes that patients are not in adrenal crisis requiring emergent treatment with high-dose corticosteroids. Glucocorticoid and mineralocorticoid treatment regimens are suggested at physiologic dosing.

# FREQUENCY OF MEASUREMENT

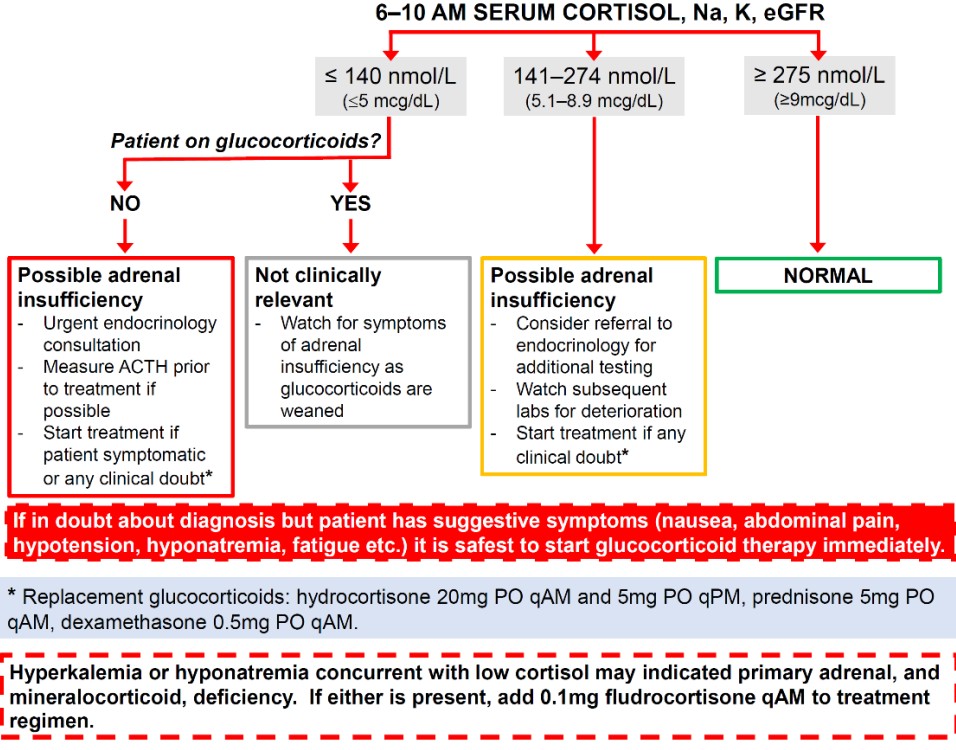

| START |
|-------|
| TREATMENT |
| STOP |

**Prior to treatment start.**

**For the duration of treatment.**
- With every cycle for the first 6 months.
- If treatment continues beyond 6 months, monthly thereafter.

**After treatment discontinuation.**
- Minimum monthly if within 12 months of treatment start.
- As clinically indicated by symptoms if >12 months out from treatment start.

# INTERPRETATION

**6–10 AM SERUM CORTISOL, Na, K, eGFR**

| ≤ 140 nmol/L (≤5 mcg/dL) | 141–274 nmol/L (5.1–8.9 mcg/dL) | ≥ 275 nmol/L (≥9mcg/dL) |
|---|---|---|

*Patient on glucocorticoids?*

**NO** / **YES**

**Possible adrenal insufficiency**
- Urgent endocrinology consultation
- Measure ACTH prior to treatment if possible
- Start treatment if patient symptomatic or any clinical doubt*

**Not clinically relevant**
- Watch for symptoms of adrenal insufficiency as glucocorticoids are weaned

**Possible adrenal insufficiency**
- Consider referral to endocrinology for additional testing
- Watch subsequent labs for deterioration
- Start treatment if any clinical doubt*

**NORMAL**

**If in doubt about diagnosis but patient has suggestive symptoms (nausea, abdominal pain, hypotension, hyponatremia, fatigue etc.) it is safest to start glucocorticoid therapy immediately.**

\* Replacement glucocorticoids: hydrocortisone 20mg PO qAM and 5mg PO qPM, prednisone 5mg PO qAM, dexamethasone 0.5mg PO qAM.

Hyperkalemia or hyponatremia concurrent with low cortisol may indicated primary adrenal, and mineralocorticoid, deficiency. If either is present, add 0.1mg fludrocortisone qAM to treatment regimen.

# For samples drawn outside of 6–10AM:

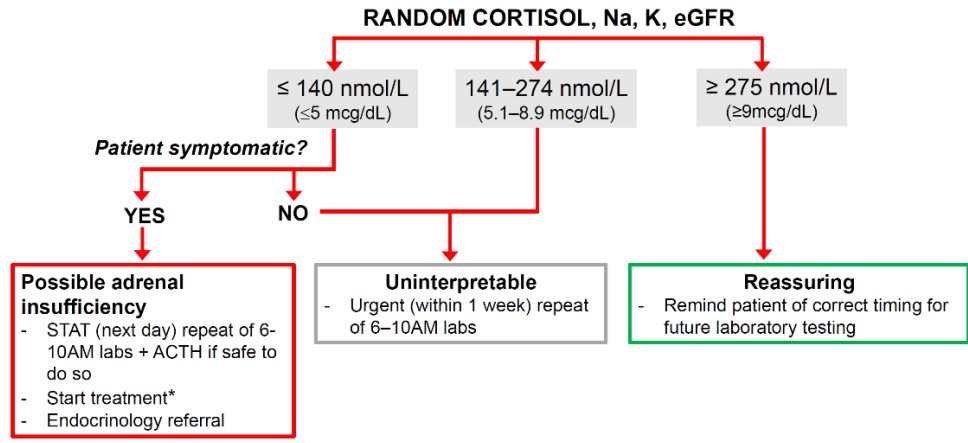

**RANDOM CORTISOL, Na, K, eGFR**

| ≤ 140 nmol/L (≤5 mcg/dL) | 141–274 nmol/L (5.1–8.9 mcg/dL) | ≥ 275 nmol/L (≥9mcg/dL) |
|---|---|---|

*Patient symptomatic?*

**YES** / **NO**

**Possible adrenal insufficiency**
- STAT (next day) repeat of 6-10AM labs + ACTH if safe to do so
- Start treatment*
- Endocrinology referral

**Uninterpretable**
- Urgent (within 1 week) repeat of 6–10AM labs

**Reassuring**
- Remind patient of correct timing for future laboratory testing

**Figure 2.** Algorithm for the frequency and interpretation of outpatient screening of asymptomatic patients for immune-checkpoint-inhibitor therapy-associated HPA-axis pathology.

Whether ICI therapy should be discontinued once diagnosis of hypophysitis is made depends on the severity of the presentation and clinical judgement. Corticosteroid therapy in this instance is used exclusively to replace endogenous function. Some sources recommend the use of high-dose corticosteroids to treat ICI-associated hypophysitis; however, it has been demonstrated that this does not reverse the loss of hormone function [40]. Mass effect with hypophysitis is rare, but if present would manifest with neurological symptoms and would necessitate high-dose glucocorticoid therapy [41]. The rates of ICI-associated primary adrenal insufficiency are very low and there are no data as to whether high-dose glucocorticoid therapy can reverse it but based on the pathophysiology of autoimmune destruction of the adrenal gland, it is unlikely.

Several other groups have proposed screening algorithms for the diagnosis of HPA-axis pathology in ICI therapy, although it is felt that none of these are as explicit and comprehensive as they lack guidance regarding the frequency and timing of monitoring, extending testing beyond treatment discontinuation, cut-offs for interpretation and action to take when cortisol is drawn outside of the accepted window [34,36,40,42–49].

## 5. Limitations

This study assessed the rates of serum cortisol measurement in the target population within one tertiary-care centre only. Patients also have blood tests done at community laboratories but those results could not be collected and the current sample is likely an underestimation of the number of tests performed within the study cohort. While this omission may be skewing the presented data on the current state of testing, the proposed screening algorithm would still apply, regardless of where patients are having their blood drawn.

The decision to measure cortisol was at the treating physician's discretion and data are not available on whether these tests were being performed for asymptomatic screening versus to confirm suspicion of HPA dysfunction.

Data on topical and inhaled glucocorticoid use were not available and may have contributed to the low measured cortisol values in patients where data were not interpretable.

A final limitation is that the proposed algorithm has not been validated and further research will be required to determine its clinical usefulness.

## 6. Conclusions

Routine cortisol measurement to diagnose ICI-HPA in asymptomatic patients should be the standard of care. The test must be drawn at a specific time and guidance on interpretation is required, but it is an easy and inexpensive test which has the potential to detect ICI-HPA in its milder forms in the outpatient setting, preventing patients from experiencing its dangerous consequences. The proposed algorithm outlines the recommended frequency of testing, including beyond treatment discontinuation, and provides specific cortisol cut-offs for interpretation and guidance on follow-up. The adoption of such an algorithm in routine practice has the potential to positively impact patients and is especially important as the number of patients receiving ICI therapy is growing.

**Author Contributions:** Conceptualization, I.D., A.I., H.L. and M.O.; methodology, I.D., J.L.V.S., H.L. and M.O.; software, J.L.V.S., A.I. and M.O.; validation, I.D. and M.O.; formal analysis, I.D., K.T.; investigation, I.D., K.T. and J.L.V.S.; resources, J.L.V.S.; data curation, I.D., K.T., J.L.V.S. and A.I.; writing—original draft preparation, I.D., K.T. and M.O.; writing—review and editing, J.L.V.S., A.I. and H.L.; visualization, I.D. and K.T.; supervision, H.L. and M.O.; project administration, A.I.; funding acquisition, N/A. All authors have read and agreed to the published version of the manuscript.

**Funding:** This research received no external funding.

**Institutional Review Board Statement:** The study was conducted in accordance with the Declaration of Helsinki and approved by the Institutional Review Board (or Ethics Committee) of The Ottawa Hospital Research Institute, (Protocol ID 5682, approved 27 October 2016).

**Informed Consent Statement:** Patient consent was waived due to that this is a retrospective research.

**Data Availability Statement:** The data presented in this study are available in this article.

**Acknowledgments:** The authors would like to thank Sara Awad for her contributions to this study.

**Conflicts of Interest:** The authors declare no conflict of interest.

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
