# Peer review of "Routine Screening for Central and Primary Adrenal Insufficiency during Immune-Checkpoint Inhibitor Therapy: An Endocrinology Perspective for Oncologists"

_curroncol, doi:10.3390/curroncol29070370_

Round 1
Reviewer 1 Report
This paper addresses the important question of whether and how to screen patients treated with immune checkpoint inhibitors for the onset of adrenal insufficiency. The authors are to be congratulated on addressing this issue. However there are some points in the manuscript which require further clarification for this to be of use to clinicians.
Methods
cortisol measurement-in-line 93 the authors need to refer to a table in which they should set out the diagnostic criteria they used to determine the aetiology of HPA axis dysfunction.
Line 105 to 106 is unclear and lead to greater clarification. It may well be that patients with HPA axis dysfunction presented with other complaints and therefore should have been included in this analysis.
Results
A discussion as to whether there were differences between those with and without cortisol measurement would be helpful with a table comparing the demographics. It would also be necessary to know how many cortisol measurements were checked specifically for screening versus because of symptoms suggestive of HPA axis dysfunction.
Table 2 requires reformatting as there are some errors in the cortisol concentrations with £ appearing. In addition the authors clarify whether the 47 points for percent of patients with concomitant glucocorticoid use included those with topical steroid or only oral?.
Line 140 to 141. This sentence requires greater clarification. A cortisol taken outside of the 6 AM to 10 AM window may still be interpretable for example if it is low in the context of acute illness.
Line 148 to 150. This sentence needs clarification. Do they mean that the 8 patients detected to have HPA axis pathology in the outpatient setting were diagnosed solely on the basis of a low screening cortisol or whether they subsequently went on to develop symptoms but could have been diagnosed on the basis of their screening test. Could the authors also show the time from starting checkpoint inhibitor to testing the cortisol and time from testing cortisol to diagnosis of HPA axis dysfunction?
Table 3 need greater clarification particularly for patients 4 and 18 who developed multiple pituitary abnormalities which has not been well described PD 1 inhibitor therapy. The patient for develop pituitary abnormalities on the CTA 4 inhibitor or whilst on the PD 1 inhibitor and could the authors comment more on the diagnosis in patient 18?
Author Response
Reviewer Comment: Cortisol measurement-in-line 93 the authors need to refer to a table in which they should set out the diagnostic criteria they used to determine the aetiology of HPA axis dysfunction.
Table 2 which summarizes this information has had a column added which identifies what criteria in the medical record were used to make the categorization and include:
- Concurrent systemic glucocorticoid use – Medical record included active prescription for systemic glucocorticoid at time of blood draw
- Hypophysitis – Patient went on to require glucocorticoid replacement and Endocrinology referral
- Pre-existing HPA dysfunction - Past medical history included pre-existing etiology of HPA dysfunction and patient was on glucocorticoid therapy prior to ICI administration
- Data uninterpretable – No record of concurrent systemic glucocorticoid use, patient did not go on to require glucocorticoid replacement
Reviewer Comment: Line 105 to 106 is unclear and lead to greater clarification. It may well be that patients with HPA axis dysfunction presented with other complaints and therefore should have been included in this analysis.
This sentence has been changed from “Demographic data as well as the nature of initial presentation” to “Demographic data as well as the location of initial presentation”. This change is intended to clarify that the information we were actually seeking was whether the patient was identified in the outpatient clinical setting or upon presenting to the Emergency Department or being admitted to hospital.
The prior sentence was also changed to “Patients with referrals for other presenting complaints were excluded, for example if the patient had new onset diabetes.”, as in its original form, the sentence had syntax error and was unclear.
ICI hypophysitis is a life-threatening condition and could not go untreated. We are confident that by assessing whether each patient with a demonstrated low cortisol went on to be referred to Endocrinology we are capturing all patients.
Results
Reviewer Comment: A discussion as to whether there were differences between those with and without cortisol measurement would be helpful with a table comparing the demographics. It would also be necessary to know how many cortisol measurements were checked specifically for screening versus because of symptoms suggestive of HPA axis dysfunction.
Patients “without” cortisol measurements may very well have had these labs done outside of our facility and are not captured. Statements to this effect have added to the results section:
“It is likely that the figure of only 265/786 patients having a cortisol measured is an under-representation as only cortisol measurements performed at our institution were captured. Many patients chose to get their routine blood work done at community laboratories, and this data was not captured. The decision to measure cortisol routinely was at the treating physician’s discretion and may be another reason for the low value.“
A similar statement is already included in the Limitations section.
Regarding the second portion of this comment as to which patients are having cortisol measured purely for screening pursues versus to follow-up on suspicion of HPA dysfunction would be extremely labor intensive as it would require review of every single clinical encounter for comments from the treating physician as to clinical symptoms etc. To reflect this deficiency the following statement has been added to the Limitations section:
“The decision to measure cortisol was at the treating physician’s discretion and data is not available on whether these tests were being done for asymptomatic screening versus to confirm suspicion of HPA dysfunction. “
Reviewer Comment: Table 2 requires reformatting as there are some errors in the cortisol concentrations with £ appearing. In addition the authors clarify whether the 47 points for percent of patients with concomitant glucocorticoid use included those with topical steroid or only oral?.
The symbol £ does not appear in our original version of the manuscript. The symbol present in Table 2 is “the less than or equal to”. It is possible that this symbol substitution was made during the upload process.
Relevant text and the table have had the word “systemic” added to glucocorticoid use to reflect the fact that we did not collected data on topical or inhaled glucocorticoid use. The following sentence was added to the limitations section to reflect the fact that we are missing this data:
“Data on topical and inhaled glucocorticoid use was not available and may have been caused low measured cortisol values in pateints where data was not interpretable.”
Reviewer Comment: Line 140 to 141. This sentence requires greater clarification. A cortisol taken outside of the 6 AM to 10 AM window may still be interpretable for example if it is low in the context of acute illness.
In accordance with this fact, this sentence has been changed to:
“The remainder of the tests, whether the result is normal or low, are not interpretable in the ambulatory care setting.”
The premise of our argument is that the cortisol values are being tested in an ambulatory asymptomatic patient. In the setting of a critically ill patient our algorithm and argument do not apply.
Reviewer Comment: Line 148 to 150. This sentence needs clarification. Do they mean that the 8 patients detected to have HPA axis pathology in the outpatient setting were diagnosed solely on the basis of a low screening cortisol or whether they subsequently went on to develop symptoms but could have been diagnosed on the basis of their screening test. Could the authors also show the time from starting checkpoint inhibitor to testing the cortisol and time from testing cortisol to diagnosis of HPA axis dysfunction?
The sentence in question has been changed to:
“Eight of those patients were diagnosed in the outpatient setting based on new symptoms consistent with hypophysitis followed by confirmatory testing, while 6 cases of HPA-axis pathology were only diagnosed after an emergency visit or hospital admission.”
Regarding the latter portion of the reviewer comment, this data is only relevant for the 8 patients having screening (as the others only had their cortisol measured AFTER the symptoms onset) so this data was pulled for only these 8 and summarized in the text, instead of the table, as follows:
“Amongst the 22 patients with HPA-axis pathology identified, all of the patients had central adrenal insufficiency and/or ACTH-deficiency. No cases of primary adrenal insufficiency were identified in our cohort. Of the 22 with ICI-HPA, 8 patients had routine cortisol measurement as part of their care prior to developing ICI-HPA and were detected to have HPA-axis pathology in the outpatient setting. The median time from initiation of treatment to first cortisol measurement in this patient group was 32 days (range 20-81), and the median time from the first cortisol measurement to diagnosis of HPA-axis dysfunction was 68 days (range 22-172).
Fourteen patients were not being screened. Eight of those patients were diagnosed in the outpatient setting based on new symptoms consistent with hypophysitis followed by confirmatory testing, while 6 cases of HPA-axis pathology were only diagnosed after an emergency visit or hospital admission. In all six of these cases, the emergency visit or admission was due to symptoms of adrenal insufficiency and not for another etiology where ICI-HPA was a secondary diagnosis. All of these patients had their first cortisol measurement after suspicious symptoms onset.”
Reviewer Comment: Table 3 need greater clarification particularly for patients 4 and 18 who developed multiple pituitary abnormalities which has not been well described PD 1 inhibitor therapy. The patient for develop pituitary abnormalities on the CTA 4 inhibitor or whilst on the PD 1 inhibitor and could the authors comment more on the diagnosis in patient 18?
With CTLA4 inhibitor therapy it has been well documented that all axes can be affected. Patients treated exclusively with PD1 primarily have their ACTH knocked out but other deficiencies have been noted in literature and this has been added to the text (with the appropriate reference) as follows:
“Post CTLA4 therapy, deficiency in all pituitary hormones has been documented. Patients who develop ICI-HPA after exclusive therapy with PD1 tend to primarily have ACTH deficiency, although other axes can be affected, such as in patient 18(16).”
Faje A, Reynolds K, Zubiri L, Lawrence D, Cohen JV, Sullivan RJ, et al. Hypophysitis secondary to nivolumab and pembrolizumab is a clinical entity distinct from ipilimumab-associated hypophysitis. Eur J Endocrinol. 2019;181(3):211-9.
Reviewer 2 Report
Druce and Colleagues provided a description of a quite wide cohort of patients at risk for ICI-induced adrenal insufficiency.
Despite several research articles and reviews have been recently published on this field, the choice to describe the clinical experience in a tertiary-care hospital can add some interesting experience on the specific field. The main limitation of this the manuscript is due to the quite low rate of inclusion in the cohort (265 out of more than 700 ICI treated patients) and the relatively low rate of serum cortisol evaluation for every patient during the study period. I suggest the Authors to discuss potential impact on these limitations on their observations.
Authors also provided a flow chart for the clinical monitoring of these patients. The flow-chart is certainly based on clinical experience of the Authors but it lacks of validation, that could represent a real novelty in this field. I suggest to discuss this point within limitations.
Finally, the manuscript is clearly written, despite a few minor point:
- Line 51-52: It seems to understand that “thyroid hormone replacement therapy in ICI induced hypothyroidism is easy and often required”. But not often subclinical hypothyroidism requires treatment and not often replacement therapy is easy to adjust. Please, rephrase this point.
- Line 66-67. This reference to thyroid hormones measurement sounds unclear. Please, rephrase.
Author Response
Reviewer Comment: Despite several research articles and reviews have been recently published on this field, the choice to describe the clinical experience in a tertiary-care hospital can add some interesting experience on the specific field. The main limitation of this the manuscript is due to the quite low rate of inclusion in the cohort (265 out of more than 700 ICI treated patients) and the relatively low rate of serum cortisol evaluation for every patient during the study period. I suggest the Authors to discuss potential impact on these limitations on their observations.
The following sentence has been added to the Limitations section:
“While this omission may be skewing the presented data on the current state of testing, the proposed screening algorithm would still apply, regardless of where patients are having their blood drawn.”
Reviewer Comment: Authors also provided a flow chart for the clinical monitoring of these patients. The flow-chart is certainly based on clinical experience of the Authors but it lacks of validation, that could represent a real novelty in this field. I suggest to discuss this point within limitations.
The following sentence has been added to the Limitations section:
“A final limitation is that the proposed algorithm has not been validated and further research will be required to determine its clinical usefulness.”
Reviewer Comment: Finally, the manuscript is clearly written, despite a few minor point: Line 51-52: It seems to understand that “thyroid hormone replacement therapy in ICI induced hypothyroidism is easy and often required”. But not often subclinical hypothyroidism requires treatment and not often replacement therapy is easy to adjust. Please, rephrase this point.
The statement has been changed to reflect this fact. It now reads:
“For example, ICI-associated thyroid dysfunction manifests most often as subclinical-thyroiditis. After a period of hyperthyroidism, the thyroiditis will often self-resolve without treatment, or can progress to permanent hypothyroidism, which is easily diagnosed biochemically and generally easily treated with thyroid hormone replacement.
Reviewer Comment: Line 66-67. This reference to thyroid hormones measurement sounds unclear. Please, rephrase.
The statement has been changed to:
“This can be contrasted with measurement and monitoring of thyroid hormones, such as thyroid stimulating hormone (TSH), which has a relatively long half-life, measured in days, and regulates more indolent processes in the body, such as basal metabolic rate and energy expenditure.”
Round 2
Reviewer 1 Report
Thank you for making these changes.
However Lines 317-320 still requires further clarification. If I have understood the new data correctly there was a median gap of two months from detecting a low cortisol to initiating glucocorticoid replacement? Is this correct? If so this suggests that screening with cortisol measurements is not really changing management.
Or do the authors mean 68 days from FIRST cortisol measurement (even if normal) to replacement? If so this is not really helpful and need to show time from first LOW cortisol in outpatient setting to starting replacement + how they excluded eg suppression from inhaled steroids before diagnosing ACTH deficiency.
This really is key to the paper as these 8 patients avoiding symptomatic presentation are the key to the entire rational of the screening proposal.
Author Response
Thank you for clarifying – I had misunderstood the prior direction. The data I presented before was indeed the time from the first cortisol measurement (even if normal) to when the diagnosis of ACTH deficiency was made.
The data you are requesting, which is much more relevant, has been re-analyzed and the time from the first LOW cortisol to when glucocorticoid therapy was initiated for the patients being screened was less than one week in all cases. Lines 317-320 have been removed and instead replaced with:
“All the patients having regular screening were started on glucocorticoid replacement therapy within one week of the first biochemical evidence of ICI-HPA. Six of the patients were started on treatment the very same day as their blood test, while the other two patients were started at one and six days after screening.”